# Developmental Milestones of Infancy and Associations with Later Childhood Neurodevelopmental Outcomes in the Adolescent Brain Cognitive Development (ABCD) Study

**DOI:** 10.3390/children9101424

**Published:** 2022-09-20

**Authors:** Haoran Zhuo, Jingyuan Xiao, Wan-Ling Tseng, Zeyan Liew

**Affiliations:** 1Department of Environmental Health Sciences, Yale School of Public Health, New Haven, CT 06510, USA; 2Yale Center for Perinatal, Pediatric and Environmental Epidemiology, Yale School of Public Health, New Haven, CT 06510, USA; 3Yale Child Study Center, Yale School of Medicine, New Haven, CT 06519, USA

**Keywords:** infancy developmental milestones, early childhood, behavioral problems, neurocognitive functions

## Abstract

The age at attaining infancy developmental milestones has been associated with later neurodevelopmental outcomes, but evidence from large and diverse samples is lacking. We investigated this by analyzing data of 5360 singleton children aged 9–10 from 17 states in the US enrolled in the Adolescent Brain Cognitive Development (ABCD) study during 2016–2020. Delays in four milestones (first roll over, unaided sitting, unaided walking, and speaking first words) were defined using the 90th percentile of age at attainment reported by children’s biological mothers. Childhood neurocognitive function was measured by research assistants using the NIH toolbox, and children reported their behavioral problems using the Brief Problem Monitor. Linear mixed-effects models were employed to investigate the association between delays in single or multiple milestones and childhood neurobehavioral outcomes. Delays in first roll over, unaided sitting, or walking were associated with poorer childhood neurocognitive function, while delay in speaking first words was associated with both poorer neurocognitive function and behavioral problems. Children who had delays in both motor and language milestones had the worst neurocognitive function and behavioral outcomes. Our results suggest that delays in motor and language milestone attainment during infancy are predictive of childhood neurobehavioral outcomes.

## 1. Introduction

Delays in the attainment of developmental milestones in infancy and toddlerhood are predictive of several neurodevelopmental disorders later in life [1,2,3,4,5]. However, prior work in the literature mostly comprises small-scale clinical studies or high-risk populations of individuals born preterm or small for gestational age [6,7,8]. Whether delays in infancy milestone attainment are predictive of subclinical deviations of neurodevelopmental function in the general population is less clear. Cohort studies that investigated the associations between early life milestone attainment and childhood neurodevelopment are sparse and mainly from European countries [9,10,11,12]. Currently, no studies have been conducted in large and diverse samples from the US population to investigate the associations between multiple infancy milestone attainment and childhood neurocognitive and behavioral outcomes.

In this study, we evaluated the associations between four motor and language developmental milestones during infancy and childhood neurocognitive function and behavioral outcomes using a geographically and socio-demographically diverse sample of US children. We examined the impacts of developmental delays in single or multiple milestones and conducted analyses, adjusting for several maternal and perinatal risk factors that influence early brain development.

## 2. Methods

### 2.1. Study Sample

We analyzed data collected from the Adolescent Brain Cognitive Development (ABCD) study, which is the largest ongoing longitudinal study of brain development and child health in the United States (details of the study design and data collection protocol can be found at https://abcdstudy.org/, accessed on 1 June 2021) [13]. Briefly, 11,878 youths from 21 sites across 17 states were enrolled at age 9−10, and the participants agreed to be followed for approximately 10 years into young adulthood. The public-school recruitment approach and a stratified probability sampling strategy were utilized to capture the sociodemographic diversity of the US population. Children with following conditions were excluded from the ABCD study at baseline recruitment (details may be found in ABCD release note 1.0): child not fluent in English; MRI contraindication (e.g., irremovable ferromagnetic implants or dental appliances, claustrophobia); gestational age < 28 weeks or birth weight < 1200 g; birth complications that resulted in hospitalization for more than 1 month; current diagnosis of schizophrenia, autism spectrum disorder, mental retardation/intellectual disability; or known alcohol/substance use disorder [14,15]. For the current study, we focus on 5360 singleton children with a baseline parental questionnaire completed by their biological mothers that included information on early-life developmental milestones. All children included also have complete data on the neurocognitive and behavioral outcome measures. A study flowchart is presented in Appendix A. Written informed consent or verbal assent for the ABCD study were obtained from all participants, and all data collection was implemented following approval from the institutional review board (IRB) at each study site. Research protocol for this project was approved by the IRB at Yale University (IRB# 2000031703).

### 2.2. Developmental Milestones in Infancy

At baseline, the Developmental History Questionnaire (DHQ) [16,17,18] that included information on early developmental milestones was completed by the child’s biological mother. Specifically, the mothers reported the age (in months) at which their child began to roll over, sitting without assistance, walking without assistance, and speak their first word. These motor and language milestones in infancy have been suggested as salient markers that parents or healthcare professionals can observe [19]. Ages of 5 months (>90th), 10 months (>95th is used because the 90th was tied with the 75th), 14 months (>90th), and 18 months (>90th) were used as the cutoffs to indicate relative delays for rolling over, sitting without assistance, speaking first words, and walking without assistance, respectively, based on the distributions in our sample. We also classified children as having delays in more than one motor milestone (rolling over, sitting without assistance, walking without assistance) and with or without a delay in language development (speaking first words).

### 2.3. Neurocognitive Function in Childhood

The neurocognitive function of children was administered with one-on-one monitoring on iPads by professional research assistants at baseline using the National Institutes of Health Toolbox (NIH-TB) Cognitive Function Battery (ages 7–17 version), a comprehensive set of nine well-validated individual tests on childhood cognitive abilities [20]. We focused on two standardized summary scores (mean = 100, SD = 15). The fluid cognition composite score, which measures attention, executive function, memory, and processing speed, reflects the cognitive capacity used in novel situations for learning and information processing. The crystallized cognition composite score, which captures the language-related aspects that highly correlate with intellectual functioning, reflects the cognitive capacity dependent on past learning experiences and skills [21,22]. Lower scores indicate poorer cognitive abilities. A moderate correlation (0.48) was found between fluid and crystallized scores, suggesting that different, unique aspects of cognition are captured by these two composite scores.

### 2.4. Behavioral Problems in Childhood

Children completed the Brief Problem Monitor Youth form (BPM-Y) that assessed their internalizing and externalizing behavioral problems at the one-year follow-up interview after baseline of the ABCD study. The BPM-Y is a psychometrically validated short version of the Child Behavior Checklist [23], and it contains 13 items that assess the internalizing and externalizing behavioral problems within the past 6 months [24,25]. Ratings of 0 to 2 (0: not true; 1: somewhat true; 2; very true) for each item were summed to yield the internalizing (6 items; score 0 to 12) and the externalizing (7 items; score 0 to 14) subscales. Higher scores indicate more behavioral problems [25]. The internalizing problems reflect a child’s emotional and internal psychological state [26], and the externalizing problems capture a child’s problem behaviors that are directed toward the external environment [27]. Moderate correlation (0.45) was observed between the internalizing and externalizing scores.

### 2.5. Covariates

For all models, we adjusted for child’s sex and several maternal characteristics collected in the ABCD baseline questionnaire, including maternal age at delivery, race/ethnicity, current marital status, highest educational level ever received, residential average household income, maternal prenatal care utilization (<14, =14, >14 times; 14 is the total number of recommended checkups, i.e., every 4 weeks from week 4 to 28, every 2 weeks from week 28 to 36, and weekly until delivery), [28] and maternal psychological problems extracted from the adult self-report (ASR) completed by the mother [29]. In addition, we evaluated several maternal reported perinatal risk factors that may influence fetal/infancy brain development [30,31,32,33,34,35,36], including maternal substance use (i.e., cannabis, tobacco, alcohol) before and after awareness of pregnancy, pregnancy complications (including pre-eclampsia, high blood pressure, diabetes, urinary tract infection, severe anemia), delivery or birth outcomes (including preterm birth, C-section, jaundice needing treatment), and breastfeeding (having breastfed the index child for at least one month).

### 2.6. Statistical Analysis

We utilized linear mixed-effects models with random intercept for study sites to estimate the mean differences and 95% CI for the neurocognitive function scores and the behavioral problem scores according to delayed attainment of each developmental milestone and delays in more than one motor milestone (with or without a language delay) using no delays as the reference. For 7% of participants with at least one missing value for covariates of maternal characteristics, we used multiple imputations to generate five complete datasets in the procedure PROC MI and then used the PROC MIANALYZE procedure in SAS to combine the results from mixed-effects models on each complete dataset [37]. We compared models with or without adjusting for the perinatal risk factors and evaluated the direction and magnitude of the changes of associations.

To address the co-occurrence of cognitive and behavioral problems, we performed a latent profile analysis (LPA) to identify the underlying subgroups of neurodevelopmental profiles characterized by potentially heterogeneous patterns of neurocognitive and behavioral assessments (fluid, crystalized, internalizing, externalizing scores) using the tidyLPA package in R [38,39]. LPA is a Gaussian finite mixture model that identifies subgroups of commonly shared patterns across multiple continuous responses [40]. We first hypothesized multiple LPA models with different numbers of underlying profiles (three to six) and identified the model with five latent profiles to be the final solution according to the criteria of good fit that has lower Akaike information criterion (AIC) and Bayesian information criterion (BIC) scores, entropy of 0.6–0.8, and non-significant *p*-value (*p* > 0.05) from the Bootstrapped likelihood ratio test (BLRT), indicating that models with more profiles are not significantly better. Next, we performed logistic regression analyses to estimate the associations between delays of infancy milestone attainment and each of the latent profile using a “typical” profile as the reference and adjusting for maternal sociodemographic characteristics and child’s sex at birth.

We also performed stratified analyses of the main outcomes by child sex (male; female) and investigated possible sex-specific associations. The interaction *p*-value of the product term of milestone attainment and child’s sex was computed for the test of heterogeneity. All statistical analyses were performed using SAS version 9.4 (SAS Institute Inc., Cary, NC, USA) and R version 3.6.0.

## 3. Results

Table 1 shows the demographic characteristics of the study sample. Male children, children of Hispanic origin, and children born in families with lower educational level and household income were more likely to have delays in the attainment of infancy milestones we assessed.

Delays in each of the measured motor milestones (first roll over, unaided sitting, unaided walking) were associated with lower neurocognitive function scores, but no association was found for behavioral problems (Figure 1). Specifically, the later attainment of first roll over (≥5 months) was associated with both lower fluid function (mean difference: −1.82, 95% CI: −2.55, −1.09) and crystallized function scores (mean difference: −0.80, 95% CI: −1.26, −0.34). Achieving first sitting without assistance at or after month 10 was associated with lower crystallized function score (mean difference: −1.05, 95% CI: −1.92, −0.19), while children who first walked without assistance ≥ 18 months had lower fluid cognition scores (mean difference: −1.35, 95% CI: −2.13, −0.57). Delays in the language milestone (speaking first words) were associated with both poorer neurocognitive functions and more behavioral problems. The median age of speaking first words was 11 months. The later attainment at the age of 14 months or later was associated with both lower fluid and crystallized function score (fluid: mean difference: −1.52, 95% CI: −2.53, −0.51; crystallized: mean difference: −1.28, 95% CI: −1.92, −0.64) and more internalizing and externalizing behavioral problems (internalizing: mean difference 0.42, 95% CI: 0.20, 0.63; externalizing: mean difference: 0.41, 95% CI: 0.21, 0.62). These associations between each developmental milestone and childhood outcomes were mostly stable, with only a few showing a moderate attenuation (a change of effect estimate < 13%) when adjusting for the list of perinatal risk factors in the analyses (Appendix A). No apparent difference was found in the sex-stratified analyses (*p*-values for heterogeneity > 0.05) (Appendix A).

When examining delays in multiple milestones, we found that the magnitude of associations for having delays in two or more motor milestones and the neurocognitive scores were larger than having only one motor milestone delay, compared with children without any delays (e.g., fluid: two or more (mean difference: −2.52, 95% CI: −4.00, −1.03), one (mean difference: −1.13, 95% CI: −1.79, −0.47)). Children who had delays in more than one motor milestone and language milestone had the worst neurocognitive scores (fluid: mean difference: −3.30, 95% CI: −5.55, −1.05; crystallized: mean difference: −2.65, 95% CI: −4.07, −1.23) and also more behavioral difficulties (internalizing: mean difference: 0.47, 95% CI: 0.01, 0.95; externalizing: mean difference: 0.41, 95% CI: −0.05, 0.87) (Figure 1 and Appendix A).

In the LPA, five distinct profiles were identified—60.6% of children were classified as “Typical behavioral and cognitive outcomes” with an average level of childhood neurocognitive and behavioral outcomes, 12.0% were classified as having “Poorer behavioral and cognitive outcomes” with at least one standard deviation from the average scores, 11.7% as having “Poorer behavioral outcomes”, 9.7% as having “Poorer cognitive outcomes”, and 6.0% children had “Better cognitive outcomes” (Figure 2a). A delay in first roll over was strongly associated with having typical behaviors but a poorer cognition (OR = 1.66, 95% CI: 1.28, 2.15), while a delay in speaking first words was associated with three profiles including poorer cognition only (OR = 1.47, 95% CI: 0.99, 2.19), poorer behaviors only (OR = 1.49, 95% CI: 1.08, 2.04), or the co-occurrence in both (OR = 1.95, 95% CI: 1.43, 2.67) (Figure 2b and Appendix A).

## 4. Discussion

We found that delays in attaining motor and language milestone before age 2 were associated with lower cognitive function and more behavioral difficulties in children aged 9–10 in the US. Specifically, a delayed age of first roll over was most strongly associated with lower cognitive function, while a delay in the language milestone (i.e., speaking first words) was associated with both poor cognitive and behavioral outcomes. Several maternal and perinatal risk factors of brain development that we examined did not explain these observed associations.

Early motor development has long been hypothesized as a prognostic factor for later neurocognitive disorders [41,42]. It has been suggested that early establishment of functional connectivity between the frontal cortex and basal ganglia may lead to early attainment of motor milestones and a subsequently better cognitive function [43,44]. However, only four cohort studies have investigated the link between developmental motor milestones and later neurocognitive abilities. Our findings in the ABCD generally corroborate these cohort studies, including a British cohort that reported delays in unaided standing and walking in infancy associated with poorer intellectual performance at ages 8, 26, and 53 years [9]. In a study of 599 children from the New York State, delays in standing with assistance were associated with lower cognitive skills at age 4 [45]. Similarly, a Northern Finland cohort found delays in standing were associated with poorer cognitive executive function in adulthood (age 33–35) [10]. Finally, the age of attaining unaided standing, walking, and sitting was negatively associated with IQ score at age 64 months in a longitudinal Bangladesh study [46].

In terms of early-life language development and later cognitive functions, our findings are consistent with the one study in a British cohort reporting that delays in speaking in infancy were associated with lower IQ at age 8 and poorer cognitive functions at ages 26 and 53 years [9]. Three other cohort studies have reported the effect of delays in language/communication attainment on the risk of emotional and behavioral problems [11,47,48]. It is well-documented that some children are affected by co-occurring language delay and neurobehavioral problems [49,50,51,52]. The underlying genetic, social and environmental risk factors that affect various neurodevelopmental domains have been proposed to explain these observed co-morbidities [53,54,55]. Alternatively, young children with language delay may experience difficulties such as frustration, low self-esteem, or social rejection that can further contribute to behavioral problems later in life [56,57].

The percentage of delays in milestone attainment in our study using the ABCD study is lower than the national estimates of development delays when applying the CDC guidelines on milestone attainment [58]. This might partly reflect errors in maternal recall or that the ABCD study has excluded children with several major neurological disorders, extreme birth outcomes, or birth complications that resulted in hospitalization for more than 1 month in the baseline enrollment [14,15]. Our study shows that a delay in milestone attainment, even when using subclinical cutoff values in a sample of children without several major neurological disorders, is predictive of childhood cognitive and behavioral functioning. Future studies that include a larger sample of infants with varying degrees of clinically confirmed developmental delays are recommended.

Our study has several strengths. Our report is based on a large and ethnically and socially diverse sample of US children, and thus the results promote generalizability. We were able to examine delays in multiple milestones and study both cognitive and behavioral outcomes using a multi-method, multi-informant approach (i.e., mother-report on milestones, child-report on behavioral outcomes, and computerized tasks for cognitive function). Another strength is that we used the latent profile models to investigate multiple neurodevelopmental outcomes. The ABCD has collected rich information on perinatal risk factors of early brain development, allowing evaluation of their influence in the observed associations. Despite these strengths, our study has limitations. First, recall bias of the developmental milestone data can be a concern in the ABCD study. However, research has suggested maternal recall on children’s key milestone achievement, especially for developmental delays, could be reliable and has been used for health studies [59,60]. While the ABCD has collected mother-reported child outcomes, we focused on studying the computerized-derived cognitive scores and child-reported behavioral outcomes to avoid strong differential and dependent exposure and outcome misclassification errors using data solely based on maternal reports [61]. Another limitation is the ABCD’s measurement of language development only measured the age of speaking the first word, which was limited to early stage of language development. Future research that examines speech, language, and communicative milestones at different developmental stages are needed. Moreover, although we evaluated influence from an array of perinatal risk factors, including maternal sociodemographic and lifestyle and obstetric factors, additional postnatal and/or early childhood familial, [62] genetic, [63] and environmental risks [64], as well as community level physical and social factors [65] could be explored in future research. Finally, our study focused on cognitive and behavioral outcomes. An assessment for other mental health endpoints (e.g., depression and anxiety symptoms) is warranted for the ABCD study when the children are followed into adolescence and young adulthood.

## 5. Conclusions

We found that delays in infancy motor and language milestone attainment before age 2 were predictive of childhood cognitive function and behavioral difficulties in a diverse sample of US children from the ABCD study. Screening for delays in infancy milestone attainments may help parents and clinicians to identify children at risk for long-term neurodevelopmental problems. Such early screening is critical for providing timely services and interventions to ensure healthy neuropsychological development.

## Figures and Tables

**Figure 1 children-09-01424-f001:**
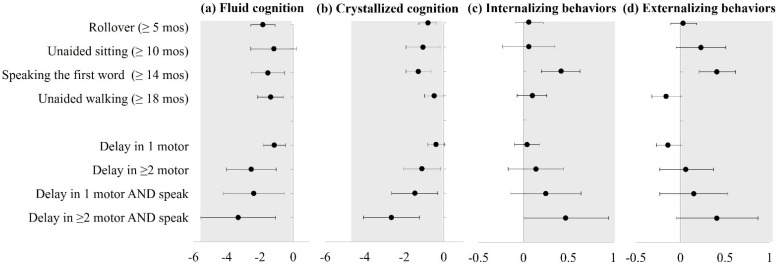
Estimated mean score differences (and 95% CI) of delays in individual milestone attainment (top panel) and multiple milestone attainment (bottom panel) with childhood neurodevelopment at ages 9–10, including: (**a**) fluid cognition; (**b**) crystallized cognition; (**c**) internalizing behaviors; (**d**) externalizing behaviors, adjusted for maternal sociodemographic characteristics (age at delivery, race/ethnicity, marriage status, education level, residential average of family income, self-reported mental health problems) and child’s sex at birth. Note: Gray area indicates poorer neurodevelopmental outcomes, with lower scores of cognition tests and higher scores of behavioral tests indicating more problems. In total, 3.1% children had delays in ≥2 motor milestones, and 1.3% children had delays in ≥2 motor and language milestone.

**Figure 2 children-09-01424-f002:**
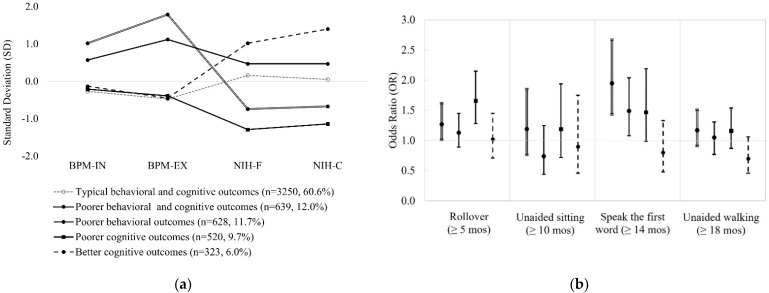
Estimated association between delays in infancy milestone attainment and LPA-identified profiles of childhood neurodevelopment outcomes at age 9–10, adjusted for maternal sociodemographic characteristics (age at delivery, race/ethnicity, marriage status, education level, residential average of family income, self-reported mental health problems) and child’s sex at birth. (**a**). Latent profiles of childhood neurodevelopment (fluid/crystallized cognition, internalizing/externalizing behaviors) at age 9–10; (**b**). The odds ratio (OR) and 95% confidence interval (CI) for latent profiles of childhood neurodevelopment at age 9–10 according to delays in infancy milestone attainment. Note: BPM-IN: internalizing behavioral scores from BPM-Y; BPM-EX: externalizing behavioral scores from BPM-Y; NIH-F: fluid cognition composite scores from NIH-TB; NIH-C: crystallized cognition composite scores from NIH-TB.

**Table 1 children-09-01424-t001:** Characteristics of the study population and children with delays in milestone attainment.

	Study Population (*N* = 5360)	Rolling over Delay (*N* = 802)	Unaided Sitting Delay (*N* = 196)	Speaking First Words Delay (*N* = 378)	Unaided Walking Delay (*N* = 676)
	%	%	%	%	%
Child’s sex at birth (Male)	52.8	55.7	63.3	66.7	54.5
**Maternal characteristics**					
Age at delivery (years; mean ± SD)	29.2 ± 6.1	29.0 ± 6.2	29.5 ± 6.3	29.9 ± 6.3	29.8 ± 6.6
Race/Ethnicity					
Non-Hispanic white	59.0	51.4	61.2	57.9	51.0
Non-Hispanic black	13.4	12.8	9.7	11.9	16.7
Spanish/Hispanic/Latino	17.9	27.8	19.9	19.8	21.6
Native American, Asian, Others	4.0	3.5	3.6	4.8	4.9
Missing/unknown	5.6	4.5	5.6	5.6	5.8
Married/living with partner	74.0	73.8	67.9	75.4	67.6
Missing/unknown	0.7	0.6	0.5	1.1	1.2
Educational level					
High school/GED or less	16.5	24.2	20.9	12.4	21.2
Some college or associate degree	29.5	28.8	31.1	30.0	32.5
Bachelor’s degree or above	53.9	46.9	48.0	57.4	46.2
Missing/unknown	0.1	0.1	-	0.3	0.2
Residential average of family income (per USD 10,000; mean ± SD)	7.6 ± 3.6	7.1 ± 3.5	7.2 ± 3.2	7.8 ± 3.6	7.3 ± 3.7
Regular prenatal care utilization ^1^	86.2	77.9	81.1	80.9	87.3
Self-report mental health problems ^2^	2.7	2.5	4.1	4.0	3.9
**Perinatal factors**					
Cannabis use before and after awareness of pregnancy	5.5	4.7	3.6	5.6	5.6
Tobacco uses before and after awareness of pregnancy	12.6	11.1	13.8	12.4	12.0
Alcohol uses before and after awareness of pregnancy	25.8	20.6	18.4	26.5	25.6
Pregnancy-related complications					
Pre-eclampsia/eclampsia/toxemia	5.6	7.5	9.7	6.4	7.3
High blood pressure	8.3	9.9	8.7	11.4	9.8
Diabetes	6.3	7.6	8.7	7.7	6.1
Urinary tract infection (UTI)	8.2	10.7	9.2	8.7	9.0
Severe anemia	4.3	5.4	4.1	5.3	4.0
Delivery-related outcomes					
Preterm birth	9.6	16.3	21.4	13.5	12.9
C-section	30.2	32.7	39.3	28.4	37.4
Jaundice needing treatment	14.7	16.7	20.4	17.2	15.2
Breastfeeding ^3^	82.0	80.8	76.5	82.5	76.3

^1^ Defined as at least 14 times in total (by having a checkup every four weeks from week 4 to 28, every two weeks from week 28 to 36, and every week from week 36 to 41). ^2.^ Evaluated by adult self-report (ASR) using clinical cutoff score T-score > 63. ^3.^ Defined as breastfeeding for at least 1 month. Note: Milestone delays are defined as: rolling over (≥5 months), unaided sitting (≥10 months), speaking first words (≥14 months), and unaided walking (≥18 months).

## Data Availability

The data used are from the ABCD study and are freely available to all institutional members who comply with the ABCD data use agreement. The list and definitions of variables collected in the database can be found at the ABCD website (https://abcdstudy.org/, accessed on 1 June 2021).

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
