# Peer review of "Developmental Milestones of Infancy and Associations with Later Childhood Neurodevelopmental Outcomes in the Adolescent Brain Cognitive Development (ABCD) Study"

_children, 2022, doi:10.3390/children9101424_

Round 1

Reviewer 1 Report

The age at which developmental milestones are reached in infancy is associated with later neurodevelopmental outcomes, but evidence from large and diverse samples is lacking. We believe it is significant that our analysis of a large data set showed that delays in first turning over, sitting on one side, and walking were associated with a neurocognitive decline in childhood and that delays in the first speech were associated with both neurocognitive decline and behavioral problems.

Introduction

It is written in a concise manner. No special comments.

Results

The percentage and distribution of children with delays in achieving milestones among the study population should be described with a check for any significant differences compared to previous reports.

Lines 166-

Is the number before the 95% confidence interval the odds ratio?

Although environmental and gender differences are highly relevant to language development, were there gender differences in the incidence of problems in this study? It is noted that cohort studies have reported that delayed language and communication skills affect the risk of emotional and behavioral problems, but were there any gender effects on these results?

Author Response

The age at which developmental milestones are reached in infancy is associated with later neurodevelopmental outcomes, but evidence from large and diverse samples is lacking. We believe it is significant that our analysis of a large data set showed that delays in first turning over, sitting on one side, and walking were associated with a neurocognitive decline in childhood and that delays in the first speech were associated with both neurocognitive decline and behavioral problems.

Thank you for reviewing our article. We have carefully considered your comments and please see our point-to-point response below. All revised contents in the manuscript are indicated in red.

  1. Introduction

It is written in a concise manner. No special comments.

Thank you for your positive feedback.

  1. Results

The percentage and distribution of children with delays in achieving milestones among the study population should be described with a check for any significant differences compared to previous reports.

The percentage of delays in milestones attainment in our study using ABCD study is lower than when applying CDC guidelines on milestone attainment.1 This might partly reflect errors in maternal recall or that the ABCD study has excluded children with several major neurological disorders, extreme birth outcomes, or birth complications that resulted in hospitalization for more than 1 month in the baseline enrollment.2, 3 Our study shows a delay in milestones attainment, even when using sub-clinical cutoff values in a sample of children without several major neurological disorders, is predictive of childhood cognitive and behavioral functioning. Future studies that include a larger sample of infants with varying degrees of clinically confirmed developmental delays are recommended.

We have added these discussions in our manuscript (see lines 57-63 and lines 268-277 for the updates).

  1. Lines 166-Is the number before the 95% confidence interval the odds ratio?

This is the estimated mean difference for the neurocognitive function score according to the delayed attainment of developmental milestones from the linear mixed-effect model. We have added the notation in the manuscript (see lines 173-185 and lines 204-209 for the updates).

  1. Although environmental and gender differences are highly relevant to language development, were there gender differences in the incidence of problems in this study? It is noted that cohort studies have reported that delayed language and communication skills affect the risk of emotional and behavioral problems, but were there any gender effects on these results?

We agree that sex differences with higher prevalence of language delays in male children have been consistently reported. Indeed, male children were more likely to have exhibit delays in their first word spoken than girls reported in the ABCD study (11.8% vs. 7.0%),2 and we have also observed the same trend. However, based on our child’s sex stratified analysis and the heterogeneity test, we did not find evidence to suggest a significant sex difference in the associations between speech developmental delays and childhood emotional/behavioral problems (supplementary table 4).

Reviewer 2 Report

Thank you so much for giving me the opportunity to review this interesting and well written paper. Generally, it is very well written and clear and also the authors are using publically available database which enable gathering large amount of data which could be interpreted in context of the whole population. I think that the manuscript is well- organized also when it comes to statistical analysis. Similarly, I appreciate the quality of figures, which are illustrative. I have only small suggestions which the authors could consider to take into consideration. Please correct me if I am wrong, but I am not sure whether there is a potential overlap between delays in physical/developmental milestones and intelectual disability which could influence the outcomes of this study - did the authors exclude these participants or adjusted for it? Maybe this should be added to limitations. When it comes to future research, it would be interesting to compare whether social deprivation influences on developmental delay and this, in turn, can have an impact on neurocognitive function. 

Author Response

Thank you so much for giving me the opportunity to review this interesting and well written paper. Generally, it is very well written and clear and also the authors are using publicly available database which enable gathering large amount of data which could be interpreted in context of the whole population. I think that the manuscript is well- organized also when it comes to statistical analysis. Similarly, I appreciate the quality of figures, which are illustrative.

Thank you for your positive feedbacks. We have carefully considered your comments and please see our point-to-point response below.

  1. I have only small suggestions which the authors could consider to take into consideration. Please correct me if I am wrong, but I am not sure whether there is a potential overlap between delays in physical/developmental milestones and intellectual disability which could influence the outcomes of this study - did the authors exclude these participants or adjusted for it? Maybe this should be added to limitations.

Children already affected several neurological disorders at baseline (9-10 years), including intellectual disability, is excluded from the ABCD study eligibility. We have added these exclusion criteria to the manuscript (lines 57-63): ABCD study excluded children with following conditions at baseline recruitment (details may be found in ABCD release note 1.0): child not fluent in English, MRI contraindication (e.g., irremovable ferromagnetic implants or dental appliances, claustrophobia), gestational age <28 weeks or birth weight <1,200 grams (g), birth complications that resulted in hospitalization for more than 1 month, or have a current diagnosis of schizophrenia, autism spectrum disorder, mental retardation/intellectual disability, or known alcohol/substance use disorder. 2, 3  

Hence, intellectual disability status of the child is unlikely to influence our finding. However, this sample cannot be used to answer whether the child’s intellectual disability status modify the associations observed between infancy milestones delay and childhood outcome.

  1. When it comes to future research, it would be interesting to compare whether social deprivation influences on developmental delay and this, in turn, can have an impact on neurocognitive function. 

Thank you for this interesting suggestion. We have added the discussion of whether postnatal and early childhood factors, including individual and community level physical, social, and environmental factors, may influence the associations between milestones delay before age 2 and the mid-childhood cognitive and behavioral functioning. Please see lines 299-301 for the updates. Identifying these early childhood modifiable risk factors would inform intervention strategies to improve long-term health for infants affected by developmental delays.

Reference

  1. Zubler JM, Wiggins LD, Macias MM, et al. Evidence-Informed Milestones for Developmental Surveillance Tools. Pediatrics. 2022;149(3)doi:10.1542/peds.2021-052138
  2. Palmer CE, Sheth C, Marshall AT, et al. A Comprehensive Overview of the Physical Health of the Adolescent Brain Cognitive Development Study Cohort at Baseline. Original Research. Frontiers in Pediatrics. 2021-October-05 2021;9doi:10.3389/fped.2021.734184
  3. Karcher NR, Barch DM, Avenevoli S, et al. Assessment of the Prodromal Questionnaire–Brief Child Version for Measurement of Self-reported Psychoticlike Experiences in Childhood. JAMA Psychiatry. 2018;75(8):853-861. doi:10.1001/jamapsychiatry.2018.1334